# Redefining the Prognostic Value of High-Sensitivity Troponin in COVID-19 Patients: The Importance of Concomitant Coronary Artery Disease

**DOI:** 10.3390/jcm9103263

**Published:** 2020-10-12

**Authors:** Marco Schiavone, Alessio Gasperetti, Massimo Mancone, Aaron V. Kaplan, Cecilia Gobbi, Giosuè Mascioli, Mattia Busana, Ardan M. Saguner, Gianfranco Mitacchione, Andrea Giacomelli, Gennaro Sardella, Maurizio Viecca, Firat Duru, Spinello Antinori, Stefano Carugo, Antonio L. Bartorelli, Claudio Tondo, Massimo Galli, Francesco Fedele, Giovanni B. Forleo

**Affiliations:** 1Department of Cardiology, ASST-Fatebenefratelli Sacco, Luigi Sacco University Hospital, 20157 Milan, Italy; marco.schiavone@unimi.it (M.S.); alessio.gasperetti93@gmail.com (A.G.); gianfrancomit@hotmail.com (G.M.); viecca.maurizio@asst-fbf-sacco.it (M.V.); forleo@me.com (G.B.F.); 2Department of Clinical Internal, Anesthesiological and Cardiovascular Science, Sapienza University of Rome, 00161 Rome, Italy; gennaro.sardella@uniroma1.it (G.S.); francesco.fedele@uniroma1.it (F.F.); 3Department of Cardiovascular Medicine, Dartmouth Hitchcock Medical Center, Lebanon, NH 03766, USA; aaron.v.kaplan@hitchcock.org; 4Department of Cardiology, ASST Santi Paolo e Carlo, San Paolo University Hospital, 20142 Milan, Italy; cecilia.gobbi@unimi.it (C.G.); stefano.carugo@unimi.it (S.C.); 5Cardiovascular Department, Humanitas Gavazzeni Hospital, 24125 Bergamo, Italy; gmascioli63@gmail.com; 6Department of Anesthesiology, Emergency and Intensive Care Medicine, University of Göttingen, 37073 Göttingen, Germany; mat.busana@gmail.com; 7Department of Cardiology, University Heart Center Zürich, 8091 Zürich, Switzerland; ardan.saguner@usz.ch (A.M.S.); firat.duru@usz.ch (F.D.); 8Department of Infectious Diseases, ASST Fatebenefratelli-Sacco, Luigi Sacco University Hospital, 20157 Milan, Italy; andrea.giacomelli@unimi.it (A.G.); spinello.antinori@unimi.it (S.A.); massimo.galli@unimi.it (M.G.); 9Luigi Sacco Department of Biomedical and Clinical Sciences DIBIC, University of Milan, 20157 Milan, Italy; antonio.bartorelli@cardiologicomonzino.it; 10Centro Cardiologico Monzino IRCCS, 20138 Milan, Italy; claudio.tondo@cardiologicomonzino.it; 11Department of Clinical Sciences and Community Health, University of Milan, 20122 Milan, Italy

**Keywords:** chronic coronary syndromes, myocardial injury, cardiac troponin, COVID-19, coronary artery disease

## Abstract

Background: Although studies assessing cardiovascular comorbidities and myocardial injury in Coronavirus disease 2019 (COVID-19) patients have been published, no reports focused on clinical outcomes of myocardial injury in patients with and without chronic coronary syndromes (CCS) are currently available. Methods: In this study, consecutive COVID-19 patients admitted to four different institutions were screened for enrolment. Patients were divided into two groups (CCS vs. no-CCS). Association with in-hospital mortality and related predictors represented the main study outcome; myocardial injury and its predictors were deemed secondary outcomes. Results: A total of 674 COVID-19 patients were enrolled, 112 (16.6%) with an established history of CCS. Myocardial injury occurred in 43.8% patients with CCS vs. 14.4% patients without CCS, as confirmed by high-sensitivity cardiac troponin (hs-cTn) elevation on admission or during hospitalization. The mortality rate in the CCS cohort was nearly three-fold higher. After adjusting for disease severity, myocardial injury resulted significantly associated with in-hospital mortality in the no-CCS group but not in CCS patients. Conclusions: Patients with CCS and COVID-19 showed high mortality rate. Myocardial injury may be a bystander in CCS patients and COVID-19, while in patients without known history of CCS, myocardial injury has a significant role in predicting poor outcomes.

## 1. Introduction

In recent times, the available body of evidence assessing the novel Coronavirus disease (COVID-19) has led to a progressive steering from a lung-centered disease paradigm in favor of a systemic disease concept. Several studies have reported the presence of an important interplay between the cardiovascular system, coagulation derangements, and COVID-19 [1,2,3,4]. The presence of myocardial injury, defined as high-sensitivity cardiac troponin (hs-cTn) elevation, was described especially among most critically ill patients with COVID-19 [5,6,7]. In these reports, older patients with acute myocardial injury suffered from more cardiovascular (CV) comorbidities and faced less favorable prognosis [8], and biomarker elevation was present also in patients without underlying obstructive coronary artery disease (CAD) [9]. Moreover, frequency of arrhythmias was noted to be higher in patients with myocardial injury [10], potentially leading to worse outcomes.

It is well accepted that respiratory infections—e.g., influenza A/B and previous severe acute respiratory syndrome (SARS) infections—are associated with CV events, cardiac arrhythmias [11], and especially acute coronary syndromes (ACS) [12,13]. A pro-inflammatory state leading to plaque instability and prothrombotic state (type 1 myocardial infarction (MI)) or a mismatch between oxygen supply and/or demand related with respiratory failure (predominantly hypoxemia) and infectious diseases (type 2 MI) have been proposed to explain those links [14,15,16]. Similar mechanisms have been postulated to explain the onset of myocardial injury in relationship with severe acute respiratory syndrome coronavirus 2 (SARS-CoV-2) infection, although a direct myocardial “myocarditis-like” effect, potentially associated with clinically relevant cardiac arrhythmias [17], has not been completely ruled out.

Patients with chronic coronary syndromes (CCS) defined according to the European guidelines [18] may be more susceptible to triggers that can lead to type 1 or 2 MI [19,20]. Although CV diseases [11,21] and myocardial injury [5] are postulated to have a role in worsening clinical outcomes in COVID-19, clear links between history of CCS, myocardial injury, and in-hospital outcomes have not been described.

The aim of this study was to evaluate clinical outcomes of CCS patients with COVID-19 and the potential mechanisms of myocardial injury in CCS and no-CCS patients with COVID-19.

## 2. Methods

### 2.1. Study Population

Consecutive confirmed cases of COVID-19 admitted to four Italian institutions (Luigi Sacco Hospital, Milan, Italy; Policlinico Umberto I, Rome, Italy; Humanitas Gavazzeni Hospital, Bergamo, Italy; and Centro Cardiologico Monzino, Milan, Italy) between 23 February 2020 and 1 May 2020 were screened for enrolment. Patients were prospectively enrolled, and data were retrospectively analyzed. A confirmed case of COVID-19 was defined by a positive result on a reverse-transcriptase-polymerase-chain-reaction (RT-PCR) assay performed on a nasopharyngeal swab, according to the World Health Organization (WHO) guidelines.

Patients with CCS were defined as specifically stated by the European Guidelines [18], as follows: “asymptomatic and symptomatic patients with stabilized symptoms <1 year after ACS or patients with recent revascularization; asymptomatic and symptomatic patients >1 year after initial diagnosis or revascularization”.

Patients presented to the emergency department (ED) with SARS-CoV-2 infection and ACS were excluded from the analysis. Clinical outcomes were monitored up to the final date of follow-up (11 May 2020). All patients still hospitalized or with an incomplete follow-up were excluded.

This study was approved by the local ethical board (project identification code 2020/ST/081) in accordance with national regulations and was conducted in accordance to the Declaration of Helsinki. The cause of death in all patients were adjudicated in a standardized fashion.

### 2.2. Data Collection

Baseline demographic, clinical, laboratory, and radiological characteristics as well as treatment and outcomes data were obtained from in-hospital electronic medical records on each patient. Myocardial injury was defined according to the European Guidelines for the fourth universal definition of MI, i.e., elevated hs-cTn with at least one value (on admission or during hospitalization) above the 99th percentile upper reference limit (URL), regardless of new electrocardiographic or echocardiographic abnormalities. Hs-cTn levels analysis was performed using peak hs-cTn values during hospitalization. Fever was defined as axillary temperature ≥37.5 °C. Acute respiratory distress syndrome (ARDS) was defined according to the Berlin definition [22]. Chronic kidney disease (CKD) was defined by a glomerular filtration rate (eGFR) ≤ 60 mL/min/m^2^, obtained with the CKD-EPI equation. Baseline laboratory values were analyzed on arterial and venous blood samples collected at the time of hospitalization. Radiological findings were assessed using a chest radiograph severity scoring system validated in patients with severe acute respiratory infection [23]. All data were independently reviewed and entered into a general and de-identified database by 4 different data collectors.

### 2.3. Cohort Definition and Study Outcome

Patients were classified into two groups (CCS and no-CCS) according to their medical history. Association with in-hospital mortality and related predictors represented the main study outcome; myocardial injury and its predictors were deemed secondary outcomes.

### 2.4. Statistical Analysis

Categorical variables are reported as counts (percentage). Normality of distribution was tested for all continuous variables using a Shapiro–Wilk test. Continuous variables are reported as mean ± standard deviation (s.d.) or as median [inter-quartile range] (IQR) if normally or nonnormally distributed, respectively. Comparisons were performed using a student T test, a Mann Whitney U test, a Chi Square test, or a Fisher exact test, as appropriate. A multivariate logistic regression was performed to test the association between dichotomic outcomes (mortality and myocardial injury) and cardiovascular risk factors and/or parameters of clinical interest for COVID-19 patients; strength of association is reported using odds ratios (OR). A by-group survival analysis was performed using Kaplan–Meier curves. Kaplan–Meier curves were compared first using a log-rank test for overall difference among the three groups, and then a post hoc all pair-wise comparisons was performed using a Bonferroni correction. A two-tailed *p* value < 0.05 was deemed as statistically significant. All statistical analyses were performed using STATA version 14.0 (Stata Corp, TX, USA).

## 3. Results

### 3.1. Baseline Characteristics

A total of 674 consecutive COVID-19 patients were prospectively enrolled, 112 (16.6%) of whom had an established history of CCS. CCS patients were significantly older than no-CCS patients and presented with more CV risk factors. Both CV (heart failure and atrial fibrillation) and non-CV (chronic obstructive pulmonary disease (COPD) and chronic kidney disease (CKD)) comorbidities were more frequent in the CCS group. Patients with CCS were more likely to receive CV medications. Baseline characteristics of the whole cohort are summarized in Table 1.

Clinical characteristics on admission are summarized in Table 2.

CCS patients were less likely to present to the ED with fever, cough, and sore throat, although no difference in the time interval between symptom onset and admission was found. Oxygen saturation (SaO_2_) and PaO_2_/FiO_2_ ratio were significantly lower among CCS patients. Patients with CCS had more severe findings on admission chest x-ray.

### 3.2. In-Hospital Outcomes

Median in-hospital stay, intensive care unit (ICU) admissions, and use of mechanical invasive ventilation were similar in the two groups. When compared to non-CCS patients, CCS patients received noninvasive ventilation (NIV) more frequently and developed ARDS more often. Myocardial injury occurred in 43.8% patients with CCS vs. 14.4% patients without CCS (*p* < 0.001), as confirmed by elevation of hs-cTn at admission or during hospitalization. The mortality rate in the CCS cohort was nearly three-fold higher.

The total data from in-hospital stay are summarized in Table 3.

### 3.3. Mortality and Outcome Predictors

At overall multivariate logistical regression, PaO_2_/FiO_2_ < 300, HF, age > 65 years, male gender, myocardial injury, and CCS were independently and significantly associated with an increased risk of death during hospitalization (Figure 1A and Appendix A).

A stratification of in-hospital mortality between CCS and non-CCS was performed and is reported in Figure 1B and Appendix AA,B. Myocardial injury was significantly associated with in-hospital mortality in the no-CCS group but not in CCS patients. A low PaO_2_/FiO_2_ ratio was a significant mortality predictor in both groups, while HF was a significant mortality predictor in non-CCS patients and reached borderline significance only in CCS patients. Of note, CCS patients without prior percutaneous coronary intervention (PCI) or coronary artery bypass graft (CABG) showed a trend towards significance for a higher risk of death. Figure 2 shows Kaplan–Meier survival curves for CCS patients, no-CCS patients, and no-CCS patients developing myocardial injury during in-hospital stay.

Both CCS and no-CCS patients with myocardial injury had a higher mortality than no-CCS patients (both < 0.001), while no difference between groups was found (log rank *p* = 0.171). A second multivariate analysis exploring predictors of myocardial injury is reported in Figure 1C,D and Appendix A.

## 4. Discussion

Although studies assessing cardiovascular comorbidities in COVID-19 patients have been published, this study, to the best of our knowledge, is the first to be specifically focused on clinical outcomes associated with myocardial injury in patients with and without CCS.

### 4.1. Outcomes in CCS Patients

Our study shows that patients with CCS and COVID-19 have a very high mortality rate (42.0%), in line with a smaller cohort of patients with CV diseases described by Inciardi et al. [21] (36%). Nevertheless, the overall mortality rate (15.6%) was similar or lower than other studies that analyzed CV involvement in COVID-19 (Shi et al. [5], 13.7%; Guo et al. [24], 23%; Zhou et al. [1], 28%; Inciardi et al. [21], 26%; and Lala et al. [7], 18.5%). However, it should be underlined that, in the Lala et al. [7] cohort, 40.1% of patients were still hospitalized. Older age was found to be an independent risk factor for increased mortality, in line with the aforementioned studies and other reports [25,26] (median age of our cohort: 60.8 ± 15.9 years). Some have postulated that elderly patients have a hyperreactive proinflammatory profile [27], especially if suffering from hypertension and diabetes, as well as a reduction in ACE-2 levels and upregulation of angiotensin II proinflammatory pathway, that may explain higher mortality rates in patients > 65 years [28,29,30].

Obviously, patients with CCS have significantly more underlying CV risk factors and more comorbidities. Several studies have shown an association between chronic CV diseases and the development of severe complications associated with COVID-19 (including ARDS and death) [5,21,24] and higher fatality rates (10.5% for those with CV diseases vs. 0.9% in patients with no comorbidities in a Chinese report) [31]. Our findings are in line with these reports.

On admission, CCS patients presented with higher creatinine and lower hemoglobin, which seem to be an expression of underlining CKD and anemia, more than multiorgan involvement due to COVID-19 severity. CCS patients were treated with similar drug regimens (hydroxychloroquine, antibiotic, antiviral, and corticosteroids). The increased mortality observed in CCS patients occurred despite a more frequent use of tocilizumab, which indeed has not shown significant impact on the clinical outcomes in another study [32], and heparin, that may represent a useful treatment when lung disease is sufficiently severe [4].

### 4.2. Myocardial Injury and Mortality in CCS Patients

Myocardial injury has been shown to be a common condition among COVID-19 hospitalized patients and has been associated with higher risk of in-hospital mortality in general populations [1,2,3,5,7,33]. Our findings are in line with these studies regarding the prevalence of myocardial injury in COVID-19 patients (19.3% in our entire cohort vs. 12% [3], 7.2% [2], 17% [1], and 19.7% [5] in other reports), apart from the Lala et al. [7] study that reported elevation of cTn in 36% of patients.

At multivariate analysis (Figure 1A), myocardial injury was an independent predictor of mortality, for which importance was overshadowed by preexisting CCS (OR 2.5 [1.3–4.5] vs. 2.3 [1.1–5.1]). Several clinical conditions causing a mismatch between oxygen supply and demand, such as an hypoxemic respiratory failure or sepsis, may induce myocardial injury and have been proposed to explain these findings [5].

Additionally, an inflammatory state may trigger production of cytokines that can activate inflammatory cells in atherosclerotic plaques [34]. Thus, it can be hypothesized that a severe SARS-CoV-2 infection may destabilize CCS and, in turn, worsening of preexisting CCS could exacerbate COVID-19 course. However, a previous report (2009) [35] showed that, in 35% of patients with SARS-CoV infection, the SARS-CoV genome was positively detected in the heart, suggesting a possible direct SARS-CoV-2 damage to cardiomyocytes [24]. All these data seem to put in evidence two possible culprit mechanisms representing the extremes of a common pathogenesis spectrum behind the development of myocardial injury in COVID-19 patients (Figure 3).

In patients with a preexisting CCS, myocardial injury may be due predominantly to a worsening of the preexisting CCS condition due to an increased mismatch of the supply/delivery of oxygen, while in patients without a preexisting diagnosis of CCS, a direct myocardial damage (myocarditis-like) may have a relevant role. However, at present, there are no findings providing evidence of direct infection and replication of SARS-CoV-2 in cardiomyocytes [14].

When evaluating myocardial injury, it should be remembered that low levels of hs-cTn (slightly above the URL) can be detected in many CCS patients [18,36] and that also mild CCS is associated with quantifiable circulating levels of hs-cTn [37]. Hence, we performed a stratified analysis of multivariate clinical data in CCS and no-CCS patients (Figure 1D). Surprisingly, our analysis showed that myocardial injury is significantly associated with in-hospital death in no-CCS patients but not in CCS patients. Several reasons may explain this finding. First, in a sizable number of CCS patients, hs-cTn elevation may be associated with preexisting CAD, more than with an acute injury triggered by a pro-inflammatory state or hypoxemia, especially when hs-cTn is only slightly above the URL. This explanation is corroborated by well-known findings in patients with stable angina, among which the presence and extent of CAD is related with circulating levels of hs-cTn, and in the absence of ischemia, suggesting an ischemia-independent mechanism of hs-cTn release [38]. Moreover, interpretation of high hs-cTn levels may be problematic in patients with CKD, who are more likely to have hs-cTn elevation in the absence of clinical myocardial ischemia. Therefore, in these patients, hs-cTn should be measured sequentially when cardiac involvement is suspected [39].

Second, a trend towards significance shown in non-revascularized patients may support our findings, since patients with moderate stenoses not suitable for revascularization or who had stable CCS treated with medications, may be more susceptible to mismatch between oxygen supply or demand and therefore may have a higher risk of death (a subanalysis was not run given the paucity of the cohort, *n* = 26). On the other hand, patients with elevated hs-cTn but without preexisting CCS have less confounders that may lead to myocardial injury, and therefore, it is more likely that hs-cTn elevation represents a new-onset myocardial involvement which can significantly worsen prognosis and clinical outcomes. Myocardial injury in no-CCS patients may be more easily detected by hs-cTn elevation in contrast to what happens in CCS patients, in whom hs-cTn elevation is more likely a nonspecific finding.

Lastly, cardiac arrhythmia is a significant cause of morbidity and mortality in critically ill COVID-19 patients [17]. Although studies specifically focused on cardiac arrhythmias in COVID-19 are lacking, myocardial involvement with arrhythmic events has been described in a non-negligible number of patients [1,40,41], especially in the most critically ill. Apart from myocardial injury, it has to been postulated that cardiac arrhythmias may also be related to potential QT-prolonging drugs, with an inherent risk of ventricular arrhythmias and torsades de point. Nevertheless, last evidences point towards a general arrhythmic safety of hydroxychloroquine, azithromycin, and other QT-prolonging drugs administered in COVID-19 [42,43]. Unfortunately, arrhythmic data were not routinely collected in this subset of patients, mostly due to logistic difficulties in continuous heart rhythm monitoring during the pandemic. Further considerations on cardiac arrhythmias and myocardial injury related to direct damage to cardiomyocytes and systemic inflammation are difficult to provide in our cohort.

### 4.3. Limitations

Our study has several limitations. First, peak hs-cTn values were used in the analysis to define myocardial injury but a number of patients did not undergo serial hs-cTn measurements. Thus, the presence of myocardial injury may be underestimated. Moreover, a subanalysis using hs-cTn as a time-dependent variable was not prespecified in the study protocol and the data needed for an unbiased analysis were not routinely collected across the several centers involved. N-terminal pro-brain natriuretic peptide (NT-pro-BNP) values were not available in the majority of patients and are not reported in our analysis. Moreover, due to the logistic limitations of managing patients in isolation, electrocardiograms (ECGs) and echocardiograms as well as arrhythmic data that were not routinely collected with ECG continuous monitoring were acquired in a few patients only.

## 5. Conclusions

Patients with CCS and COVID-19 have a poor prognosis and a high mortality rate that was nearly three-fold higher than that observed in those without a known history of CCS. Although myocardial injury has been shown to be associated with worse outcomes in COVID-19, hs-cTn elevation had a prognostic value in non-CCS patients, while it was not predictive of poor outcomes in patients with CCS. Myocardial injury may be a bystander in CCS, whereas in patients without known history of CCS, it has a significant role in predicting poor outcomes. Further data are needed to understand proper mechanisms of myocardial injury in CCS and no-CCS patients with COVID-19.

## Figures and Tables

**Figure 1 jcm-09-03263-f001:**
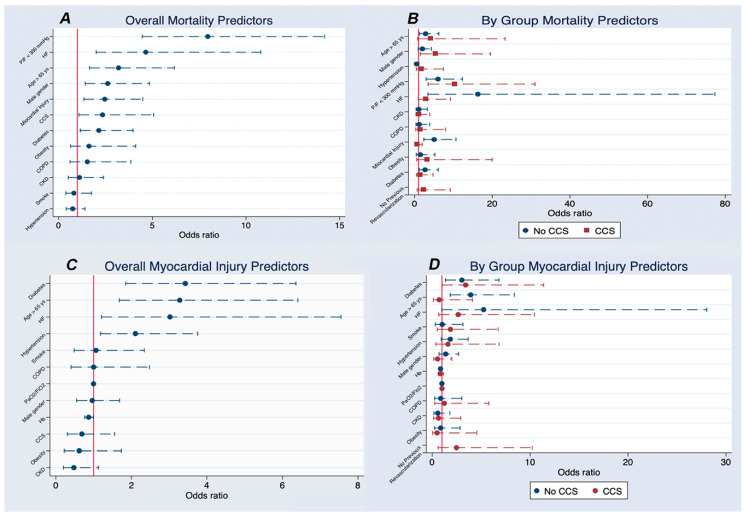
shows multivariate logistical regression analysis. Panel A: overall multivariate logistical regression for mortality. Panel B: stratification of in-hospital mortality between CCS and non-CCS patients. Panel C: overall multivariate logistical regression for myocardial injury. Panel D: stratification of myocardial injury between CCS and non-CCS patients. **Abbreviations**: CCS: chronic coronary syndromes; CKD: chronic kidney disease; COPD: chronic obstructive pulmonary disease; Hb: hemoglobin; HF: heart failure; P/F: PaO_2_/FiO_2_ ratio; ys: years.

**Figure 2 jcm-09-03263-f002:**
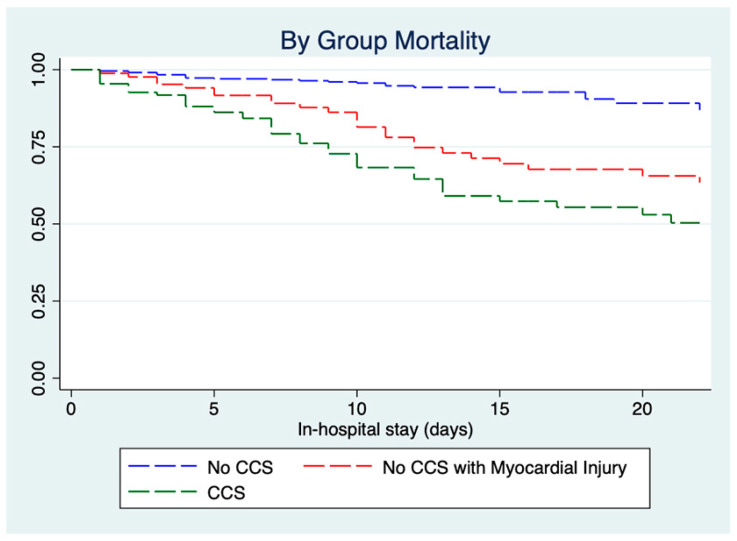
Kaplan–Meier survival curves for CCS patients, no-CCS patients, and no-CCS patients developing myocardial injury during in-hospital stay. **Abbreviations**: CCS: chronic coronary syndromes.

**Figure 3 jcm-09-03263-f003:**
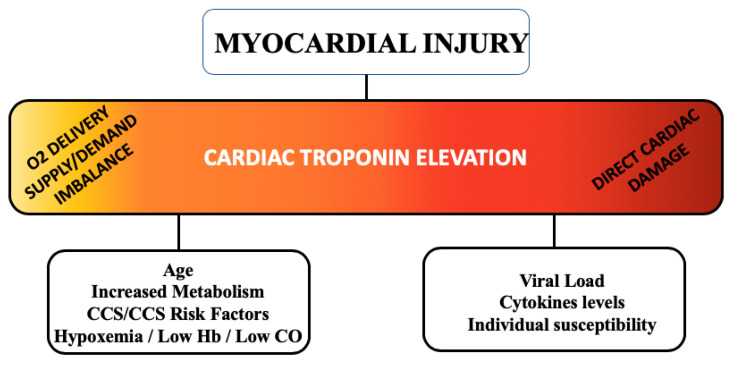
culprit mechanisms behind the development of myocardial injury in COVID-19. **Abbreviations**: CCS: chronic coronary syndromes; CO: cardiac output; Hb: hemoglobin.

**Table 1 jcm-09-03263-t001:** Baseline characteristics of the study cohort.

	Overall (*n* = 674)	CCS (*n* = 112)	No-CCS (*n* = 562)	*p*
**Age (years), mean ± s.d.**	60.8 ± 15.9	75.3 ± 10.1	58.0 ± 15.2	**<0.001**
**Sex (male), *n* (%)**	406 (60.2)	75 (67)	331 (58.9)	0.111
**Main cardiovascular risk factors, *n* (%)**				
*Hypertension*	280 (41.5)	89 (79.5)	191 (31)	**<0.001**
*Diabetes*	101 (15.0)	40 (35.7)	61 (10.8)	**<0.001**
*Dyslipidemia*	145 (21.5)	64 (57.1)	81 (14.4)	**<0.001**
*Smoke*	130 (19.3)	71 (63.4)	59 (10.5)	**<0.001**
**Comorbidities, *n* (%)**				
*Heart failure*	111 (16.5)	76 (67.9)	35 (6.2)	**<0.001**
*History of atrial fibrillation*	53 (7.9)	26 (23.2)	27 (4.8)	**<0.001**
*Chronic kidney disease*	85 (12.3)	26 (23.1)	59 (10.5)	**<0.001**
*Chronic obstructive pulmonary disease*	46 (6.8)	16 (14.3)	30 (5.3)	**<0.001**
*Cancer*	43 (6.4)	15 (13.4)	28 (5)	**<0.001**
**Drug therapy, *n* (%)**				
*ACE-inhibitors*	102 (15.2)	44 (39.3)	58 (10.3)	**<0.001**
*ARBs*	111 (16.5)	45 (40.2)	66 (11.7)	**<0.001**
*Beta-blockers*	150 (22.3)	78 (69.6)	72 (12.8)	**<0.001**
*Calcium-antagonists*	73 (10.8)	22 (19.6)	51 (9.1)	**<0.001**
*Diuretics*	96 (14.2)	46 (41.1)	50 (8.9)	**<0.001**
*VKAs*	14 (2.1)	7 (6.2)	7 (1.3)	**<0.001**
*DOACs*	30 (4.4)	11 (9.8)	19 (3.4)	**0.003**
*Antiplatelets*	130 (19.3)	108 (96.4)	22 (3.9)	**<0.001**
*Statins*	156 (23.1)	93 (83.0)	63 (11.2)	**<0.001**
Prior revascularization (PCI and/or CABG), *n* (%)	86 (76.8)	86 (78.6)	NA	**NA**

**Abbreviations**: ACE: angiotensin-converting enzyme; ARBs: angiotensin II receptor blockers; CABG: coronary artery bypass graft; CCS: chronic coronary syndromes; DOAC: direct anticoagulants; NA: not available; PCI: percutaneous coronary intervention; s.d.: standard deviation; VKAs: vitamin k antagonists. Bold text indicates a statistically significant difference with a *p*-value <0.05.

**Table 2 jcm-09-03263-t002:** Characteristics on admission.

	Overall (*n* = 674)	CCS (*n* = 112)	No-CCS (*n* = 562)	*p*
Symptoms, *n* (%)				
*Fever*	545 (80.9)	68 (60.7)	477 (84.9)	**<0.001**
*Cough*	500 (74.2)	64 (57.1)	436 (77.6)	**<0.001**
*Shortness of breath*	307 (45.5)	58 (51.8)	249 (44.3)	0.147
*Sore throat*	149 (22.1)	11 (9.8)	138 (24.6)	**<0.001**
*Myalgia*	96 (14.2)	12 (10.7)	84 (14.9)	0.242
Laboratory findings, median [IQR]				
*WBC (10^9^/L)*	5.91 [4.55–8.15]	6.7 [4.98–10.2]	5.8 [4.41–7.85]	**<0.001**
*Neutrophils (10^3^/μL)*	4.1 [2.89–6.18]	5 [3.4–7.49]	4.01 [2.81–6]	**<0.001**
*Lymphocytes (10^3^/μL)*	1 [0.70–1.42]	0.8 [0.6–1.21]	1.05 [0.74–1.46]	**<0.001**
*Monocytes (10^3^/μL)*	0.43 [0.3–0.63]	0.49 [0.3–0.7]	0.42 [0.29–0.62]	0.056
*Hb (g/dl)*	13.8 [12.8–14.8]	13 [11.5–14.1]	13.9 [13–14.9]	**<0.001**
*Platelets (10^9^/L)*	195 [154–259]	197 [148–282]	195 [154–255]	0.809
*Creatinine (mg/dl)*	0.93 [0.78–1.13]	1.1 [0.9–1.6]	0.9 [0.76–1.1]	**<0.001**
*Sodium (mmol/L)*	138 [136–140]	139 [135–141]	138 [136–140]	0.09
*Potassium (mmol/L)*	3.9 [3.6–4.2]	3.9 [3.6–4.3]	3.8 [3.5–4-2]	**0.04**
*CK (U/L)*	101 [60–187]	98.5 [48–237]	101 [61–181]	0.698
*LDH (U/L)*	311 [239–411]	355 [286–490]	304 [235–389]	**<0.001**
*ALT (U/L)*	29 [19–48]	29 [19–54]	29 [20–47]	0.817
*CRP (mg/L)*	25.4 [6.7–92.5]	15.9 [7.5–50]	30 [6.5–100]	0.075
Chest radiograph severity scoring system (Taylor et. al), median [IQR]	2 [2–3]	3 [2–4]	2 [2–3]	**<0.001**
Symptom onset to admission (days), median [IQR]	7 [3–10]	7 [3–10]	7 [3–10]	0.98
Vital status				
*Heart rate (bpm), median [IQR]*	88 [75–100]	78 [70–90]	88 [78–100]	**<0.001**
*Respiratory rate (rpm), median [IQR]*	20 [20–22]	20 [18–20]	20 [20–23]	**<0.001**
*SBP (mmHg), mean* *± sd*	130 ± 20	133 ± 22	130 ± 18	0.133
*DBP (mmHg), mean* *± sd*	77 ± 12	76 ± 14	78 ± 12	0.097
*MBP (mmHg), mean* *± sd*	96 ± 19	101 ± 37	95 ± 13	**0.002**
Arterial blood gas analysis				
*SaO_2_ (%), mean* *± sd*	95 ± 1	93 ± 1	95 ± 1	**<0.001**
*pH, mean* *± sd*	7.45 ± 0.05	7.45 ± 0.05	7.46 ± 0.05	0.434
*PaO_2_ (mmHg), median [IQR]*	77 [65–90]	77 [62–89]	77 [65–91]	0.453
*PaCO_2_ (mmHg), median [IQR]*	34 [31–37]	33 [31–36]	34 [31–37]	0.573
*PaO_2_/FiO_2_, median [IQR]*	329 [248–395]	248 [136–342]	333 [266–395]	**<0.001**

**Abbreviations**: ALT: alanine aminotransferase; CCS: chronic coronary syndromes; CK: creatine kinase; CRP: C-reactive protein; DBP: diastolic blood pressure; FiO_2_: fraction of inspired oxygen; Hb: hemoglobin; IQR: interquartile range; LDH: lactate dehydrogenase; MBP: mean blood pressures; NA: not available; PaCO_2_: partial pressure of carbon dioxide in arterial blood; PaO_2_: partial pressure of oxygen in arterial blood; s.d.: standard deviation; SaO_2_: oxygen saturation; SBP: systolic blood pressure; WBC: white blood cells. Bold text indicates a statistically significant difference with a *p*-value <0.05.

**Table 3 jcm-09-03263-t003:** Drug therapy and clinical outcomes during hospital stay.

	Overall (*n* = 674)	CCS (*n* = 112)	No-CCS (*n* = 562)	*p*
Drug therapy, *n* (%)				
*Antibiotics*	327 (48.5)	58 (51.8)	269 (47.9)	0.448
*Antivirals*	367 (54.4)	56 (50)	311 (55.3)	0.3
*Steroids*	87 (12.9)	16 (14.3)	71 (12.6)	0.634
*Hydroxychloroquine*	520 (77.1)	86 (76.8)	434 (77.2)	0.92
*Tocilizumab*	126 (18.7)	13 (11.6)	113 (20.1)	**0.035**
*Heparin*	294 (43.6)	67 (59.8)	227 (40.4)	**<0.001**
Deaths, *n* (%)	105(15.6)	47 (42.0)	58 (10.3)	**<0.001**
Intensive care unit admission, *n* (%)	46 (6.8)	6 (5.4)	40 (7.1)	0.5
Mechanical ventilation, *n* (%)	42 (6.2)	6 (5.4)	36 (6.4)	0.675
Non-invasive ventilation, *n* (%)	207 (30.7)	49 (43.8)	158 (28.1)	**<0.001**
Acute respiratory distress syndrome, *n* (%)	99 (14.7)	46 (41.1)	53 (9.4)	**<0.001**
Hospital length of stay (days), median [IQR]	10 [6–16]	11 [7–17]	10 [5–16]	0.189
Myocardial injury, *n* (%)	130 (19.3)	49 (43.8)	81 (14.4)	**<0.001**
Cardiac troponin (ng/L), median [IQR]	18 [8–40]	42 [21–75]	13 [6–27]	**<0.001**

**Abbreviations**: CCS: chronic coronary syndromes; IQR: interquartile range. Bold text indicates a statistically significant difference with a *p*-value <0.05.

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
