# Peer review of "Redefining the Prognostic Value of High-Sensitivity Troponin in COVID-19 Patients: The Importance of Concomitant Coronary Artery Disease"

_jcm, 2020, doi:10.3390/jcm9103263_

Round 1

Reviewer 1 Report

General

M. Schiavone et al has made an interesting article exploring the relationship between myocardial injury defined as hscTn > 99th percentile and Covid19 disease. It adds to existing literature with a sub-study analysis of patients with and without chronic coronary syndromes (CCS). This study highlights the troublesome nature of cTn. Even though it is an excellent marker of cardiac injury in MI, several other conditions also increases levels of cTn. As is the case in patients with Covid 19 elevation of cTn is linked to a poorer prognosis compared to those with normal levels, this has been investigated earlier. This study adds to earlier findings exploring the nature of cTn-release. Is it related to ischemia or is it related to a more general inflammatory response? As explained by the authors it seems to be two different pathways, one ischemic and one inflammatory. This might explain why cTn is linked to a poorer prognosis only in patients without CCS as this situation probably reflects a higher level of inflammation.

I think this study has some interesting results, but it needs revision. As for now there are too many spelling mistakes and formatting errors. There is also a need for a more concise and precise language. The discussion is the best written part but the other parts (including tables) need more work. I have some concerns:

Was there a difference between the groups as to when peak cTn value was achieved? As the authors have highlighted themselves cTn in relation to a general inflammation is linked to a poorer prognosis, but how can you justify to compare the groups if some patients only have a cTn upon admission and patients that dies has a peak cTn during end stage disease? Is it possible to report day of achieved peak cTn.

Some minor comments:

Abstract

Please check formatting. Line 44: Is it correct with a capital I in institutions?

Introduction.

Line 78, I understand the meaning of “high well characterized” but I do not think this is an appropriate combination of these words.

Methods:

Line 91: This is a direct transcript of the European guidelines and this should be shown.

Results:

I think it is hard to read the results since all results are referred in both text and table. If the numbers can be read in the table it is not necessary to write them in the text. Please fix for the entire results section. This will make it easier to read. Example Line 165: When compared to non-CCS patients, CCS patients received non-invasive ventilation (NIV) more frequently (43.8% vs 28.1%, p<0.001) and
developed ARDS more often (43.8% vs 28.1%, p<0.001)” could be changed to “When compared to non-CCS patients, CCS
patients received non-invasive ventilation (NIV) more frequently and
developed ARDS more often”.

Line 140: I think % is lacking in COPD, also the p value lacks a number. This can be corrected by removing the numbers all together.

Tables:

Please check formatting. Table 2 looks like it is a part of Table 1. It is preferrable if all tables are formatted in the same manner. It seems tidier to have name of all variables on the same vertical line and not aligned in the middle.

Discussion

Line 197: Should possibly be “to the best” not “to be best”

Reviewer 2 Report

In their article, "Redefining the prognostic value of high-sensitivity troponin in COVID-19 patients: the importance of concomitant coronary artery disease.", Schiavone et al. set out to assess the prognostic value of cardiac troponin as a marker of cardiac injury in COVID-19 patients. They conclude that patients with cardiovascular disease and COVID-19 had high mortality rates and myocardial injury predicts poor outcomes. The study is, in principle, quite interesting, however, there are already studies in the literature reducing the novelty of the findings reported in this paper. I have some comments.

-Why do the authors think that the median in-hospital stay, ICU admissions and use of mechanical invasive ventilation were similar in the two studied groups? Please discuss.

-In table 3, why is there less ICU admissions in CCS group? Does this mean that some patients with ARDS did not require ICU admission? Please explain.

-Was there significant arrhythmias noted in this cohort of patients?

-Arrhythmia is a significant cause of morbidity and mortality in critically ill COVID-19 patients. Please refer to the studies below and discuss them in the introduction and discussion regarding  cardiac injury and arrhythmias:
https://doi.org/10.1007/s42399-020-00454-2

-It would make it easier for the reader to compare the groups if the authors include the characteristics of patients with "no-CCS and no myocardial injury" in the tables.

Reviewer 3 Report

This is an interesting and well-written study that explores the prognostic value of high-sensitivity troponin (hs-cTn) in COVID-19 patients and the importance of concomitant coronary artery disease (CCS).

The authors conclude that patients with CCS and COVID-19 showed high mortality rate. Myocardial injury may be a bystander in CCS patients and COVID-19, while in patients without known history of CCS, myocardial injury has a significant role in predicting poor outcomes.

Although the study has certain limitations as reflected by the authors: a number of patients did not undergo serial hs-cTn measurements, the NT-pro-BNP values were not available in the majority of patients and ECG and echocardiograms were acquired only in few patients; it improves the information in the important topic of cardiovascular complications in COVID-19 patients.

Round 2

Reviewer 1 Report

Thank you for your revised version and answers to my questions. I have no further concerns regarding this paper.

Reviewer 2 Report

I thank the authors for the revision. No further comments.